# Generation of a Pure Culture of Neuron-like Cells with a Glutamatergic Phenotype from Mouse Astrocytes

**DOI:** 10.3390/biomedicines10040928

**Published:** 2022-04-18

**Authors:** Gary Stanley Fernandes, Rishabh Deo Singh, Kyeong Kyu Kim

**Affiliations:** Department of Precision Medicine, Graduate School of Basic Medical Science (GSBMS), Institute for Antimicrobial Resistance Research and Therapeutics, Sungkyunkwan University School of Medicine, Suwon 16419, Korea; garyferns@gmail.com (G.S.F.); singh.d.rishabh@gmail.com (R.D.S.)

**Keywords:** neuronal reprogramming, small molecule, astrocyte, neurons

## Abstract

Astrocyte-to-neuron reprogramming is a promising therapeutic approach for treatment of neurodegenerative diseases. The use of small molecules as an alternative to the virus-mediated ectopic expression of lineage-specific transcription factors negates the tumorigenic risk associated with viral genetic manipulation and uncontrolled differentiation of stem cells. However, because previously developed methods for small-molecule reprogramming of astrocytes to neurons are multistep, complex, and lengthy, their applications in biomedicine, including clinical treatment, are limited. Therefore, our objective in this study was to develop a novel chemical-based approach to the cellular reprogramming of astrocytes into neurons with high efficiency and low complexity. To accomplish that, we used C8-D1a, a mouse astrocyte cell line, to assess the role of small molecules in reprogramming protocols that otherwise suffer from inconsistencies caused by variations in donor of the primary cell. We developed a new protocol by which a chemical mixture formulated with Y26732, DAPT, RepSox, CHIR99021, ruxolitinib, and SAG rapidly and efficiently induced the neural reprogramming of astrocytes in four days, with a conversion efficiency of 82 ± 6%. Upon exposure to the maturation medium, those reprogrammed cells acquired a glutaminergic phenotype over the next eleven days. We also demonstrated the neuronal functionality of the induced cells by confirming KCL-induced calcium flux.

## 1. Introduction

Neurodegenerative diseases of the brain, such as Alzheimer’s and Parkinson’s, primarily manifest as a progressive loss of neurons. Similarly, traumatic brain injury causes significant neuronal and glial cell death, as a result of blunt force [1]. As the regenerative agents of the brain, neural progenitor cells are largely limited to the sub-ventricular zone and the hippocampus [2]. Interestingly, neuronal progenitor cells not only show a limited capacity to regenerate new neurons, but under specific conditions develop a propensity to differentiate into glial cells, further hindering neuronal regeneration [2,3]. Therefore, transplantation of fully differentiated neurons and direct in vivo reprogramming have emerged as strategies for the treatment of neuronal diseases [4,5]. The application of neurons differentiated from pluripotent stem cells, both embryonic and induced pluripotent stem cells, is limited by the risk of teratoma formation. Therefore, cellular reprogramming, directly converting somatic cells into neurons without going through the intermediate stages of development, has been introduced [6,7].

Astrocytes are the largest population of cells in the mammalian brain, where they provide lifelong homeostatic support to neurons [8]. Interestingly, astrogliosis, an inflammatory response seen after brain injury and in some neurodegenerative diseases, involves the activation and proliferation of astrocytes and their eventual occupation of injured brain areas [9,10,11]. These reactive astrocytes have emerged as an attractive target for direct in vivo reprogramming. Indeed, recent progress in the in situ conversion of brain astrocytes to neurons has shed light on the importance of these endogenous cell populations for treating neurodegenerative diseases [12]. In this regard, initial principal studies have successfully demonstrated the neuronal conversion of both human and mouse astrocytes to neurons or neuroblasts in vitro [13,14,15,16]. Employing the in vivo ectopic expression of pluripotent transcription factors, such as Sox2 on astrocytes, successfully generates a pool of progenitor cells, which then undergo neurogenesis [17,18,19,20]. On the other hand, the more attractive and preferable approach is direct reprogramming using the ectopic expression of the neuronal transcription factor NeuroD1, which sufficiently generates neurons without the appearance of a progenitor-like state [21,22,23]. Interestingly, employing similar investigative strategies on resident glial cells has yielded similar neurogenic potential as that of astrocytes [24,25,26]. In summary, in vivo reprogramming of glial and astroglial cells appears to be a promising strategy in regenerative medicine, opening avenues for further optimization under neurodegenerative physiological conditions.

To bypass any possible side effects associated with transcription factor-mediated re-programming, the use of small molecules is increasingly popular [6,27,28,29]. This is due to their target-specific effects, and the ease of controlling this process. Moreover, reports suggest that small molecules weighing under 500 Da show significant diffusion across the blood brain barrier, although this also depends on other factors like polar surface area, hydrogen bond, lipophilicity, and charge [30,31,32]. The first successful demonstration of the potential use of small molecules replaced existing Yamanaka factors, Oct4, Sox2, Klf4, and c-Myc, which are needed to convert somatic cells into iPS cells [33,34,35]. Similarly, the neuronal transcription factors Brn2, Ascl1, and Myt1, which were shown to drive neuronal reprogramming in fibroblasts [36] and other somatic cells, have been replaced by various chemical cocktails [37,38,39,40]. Those small molecules are responsible for modulating signaling pathways and modifying the epigenetic landscape, a requisite step for cell reprogramming, and eventually induce the expression of the transcription factors needed for a specific cell lineage [38,41].

Numerous researchers have successfully used small molecules to directly convert somatic cells, such as astrocytes and fibroblasts, into neurons. However, when astrocytes are the source, most protocols take more than two weeks and often require complex, multistep chemical cocktail treatments, making the process tedious [4,42,43,44]. Additionally, the generation of specific subtypes of neurons using small-molecule cocktails remains a challenge. Moreover, the reprogramming potential of primary astrocytes is affected by age-related gene expression changes and astrocyte heterogeneity, which are important for achieving reproducible results [45,46]. Physiological aging in humans is accompanied by chronic, low-grade, and systemic inflammation [47]. Additionally, the gene expression undergoes a complex shift that is reflected in age-related synaptic loss and neuroinflammation [48]. Furthermore, astrocytes show markedly distinct morphological, functional, and molecular variations in different spatial areas of the brain [49].

To address these drawbacks of using astrocytes and current reprogramming protocols, we developed a simple and efficient neuronal reprogramming protocol using a mouse astrocyte C8-D1a cell line, and we validated our method by confirming the functionality of the converted neuron-like cells with a glutamatergic phenotype. Our initial screening revealed that the JAK/STAT inhibitor ruxolitinib could effectively generate Tuj1^+^ cells, when used in combination with the small molecules forskolin, LDN193189, CHIR99021, and RepSox. With further optimization, we developed a neuronal reprogramming protocol that could induce a neuronal lineage in four days, with a high efficiency of 82 ± 6%. The maturation of programmed cells produced neuron-like cells with a glutamatergic phenotype. We also confirmed that our protocol successfully generated a pure culture of neuron-like cells with the properties of glutaminergic neurons.

## 2. Materials and Methods

### 2.1. Cell and Culture Conditions

The mouse astrocyte C8-D1a cell line was obtained from the American Type Culture Collection (ATCC, www.atcc.org, accessed on 27 June 2019) and maintained according to the ATCC mammalian tissue culture protocol and sterile techniques. The primary human cortical astrocytes were purchased from Thermofisher (Catalog No. N7805100) and maintained as per the standard protocol provided by the supplement.

### 2.2. Pre-Treatment of Coverslips in Cell Culture

The glass coverslips were pre-treated, as described in Richer et al. [50]. In brief, the glass coverslips were washed with detergent, dehydrated with 95% ethanol, and acid treated with HNO_3_ overnight. Before use, they were sterilized with 70% ethanol and UV light.

### 2.3. Preparation of Coverslips

The coverslips were coated with poly-l-ornithine and fibronectin, as described in Richer et al. [29]. Briefly, the coverslips were incubated in poly-l-ornithine (Sigma-Aldrich, Inc., Saint Louis, MO, USA) overnight and then treated overnight with 2 μg/mL of fibronectin (Sigma-Aldrich, Inc., Saint Louis, MO, USA).

### 2.4. Chemical Screening

A screening of small molecules from the MedChemExpress stem cell signaling library (catalogue No. HY-LO17) was performed using a basic cocktail containing forskolin (5 μM), (MedChemExpress, Monmouth Junction, NJ, USA), CHIR99021 (2 μM), (MedChemExpress, Monmouth Junction, NJ, USA), RepSox (2 μM) (MedChemExpress Monmouth Junction, NJ, USA), and LDN193189 (0.5 μM), (MedChemExpress, Monmouth Junction, NJ, USA) in Dulbecco’s modified Eagle medium: F12 (DMEM: F12) (Gibco, Thermo Fisher Scientific, Inc., Waltham, MA, USA) containing N2 (Invitrogen, Thermo Fisher Scientific, Inc., Waltham, MA, USA) as the only supplement. Each of the small molecules from the library was added to the basic cocktail at a final concentration of 5 μM and investigated for its ability to generate a neuronal-like morphology and Tuj1^+^ cells.

### 2.5. Protocol for Converting Mouse Astrocytes into Neurons

The mouse astrocytes were cultured on poly-L-ornithine- and fibronectin-coated glass coverslips (Thermo Fisher, Waltham, MA, USA) one day before neuronal induction in DMEM (Gibco, Thermo Fisher Scientific, Inc., Waltham, MA, USA), 10% Fetal Bovine Serum (Gibco, Thermo Fisher Scientific, Inc., Waltham, MA, USA), and 1% penicillin/streptomycin (Welgene, Inc., Gyeongsan, Korea). The next day, the medium was changed to the neuronal induction medium (NIM), which contained Neurobasal Plus (Life Technologies, Thermo Fisher, Waltham, MA, USA), 5 μM DAPT (MedChemExpress, Monmouth Junction, NJ, USA), 5 μM ruxolitinib (MedChemExpress, Monmouth Junction, NJ, USA), 2 μM CHIR99021, 4 μM RepSox, 200 nM SAG (MedChemExpress, Monmouth Junction, NJ, USA), and 10 μM Y26732-HCL (MedChemExpress, Monmouth Junction, NJ, USA). The NIM was supplemented with 20 ng/mL of brain-derived neurotrophic factor (BDNF) (PeproTech, Cranbury, NJ, USA), 20 ng/mL of Glial Cell Line-Derived Neurotrophic Factor (GDNF) (PeproTech, Cranbury, NJ, USA), and 20 ng/mL of Neurotrophin-3 (NT3) (PeproTech, Cranbury, NJ, USA). The mouse astrocytes were maintained in the induction medium for four days, with medium changes every two days. After four days, the induced neurons were allowed to mature in neuronal maturation medium containing Neurobasal Plus (Life Technologies, Thermo Fisher, Waltham, MA, USA) supplemented with 1% B27 (Invitrogen, Thermo Fisher, Waltham, MA, USA), 2 μM CHIR99021, 5 μM Y26732-HCL, and 200 nM Smoothened Agonist (SAG). The neuronal maturation medium was further supplemented with 100 μM dbCAMP (Sigma-Aldrich, Inc., Saint Louis, MO, USA), 20 ng/mL of BDNF, 20 ng/mL of GDNF, 20 ng/mL of NT3, and 1% penicillin/streptomycin (P/S, Welgene, Inc., Gyeongsan, Korea). The cells were kept in the maturation medium indefinitely, with medium changes every two to three days.

### 2.6. Conversion Efficiency and Neuronal Purity

The conversion efficiency was calculated as previously described [36]. Briefly, we randomly selected 5–10 view fields for each sample on day four post-induction, using an Olympus IX-71 microscope (IX71S1F3, Olympus, Tokyo, Japan), and counted the total number of Tuj1^+^ cells with neuronal morphology. The conversion efficiency was calculated as the ratio of Tuj1^+^/DAPI cells to the initial number of seeded cells in each visual field. Quantitative data are presented as the mean ± SEM of at least three independent experiments. The neuronal purity represents the percentage of induced DCX^+^/Tuj1^+^ cells in total cells stained by DAPI.

### 2.7. RNA Isolation and cDNA Synthesis

The cells were harvested from three wells of 24-well plates at the indicated time points, and the total RNA was extracted using RNeasy Mini Kit (QIAGEN, Hilden, Germany), for a total of 50–100 ng/μL of pure RNA. The isolated RNA had an A_260_/A_280_ ratio between 1.8 and 2.1, which indicates RNA purity. The isolated RNA was stored at −80°C. For cDNA synthesis, a OneScript^®^ cDNA Synthesis Kit (abm, Vancouver, Canada) was used to convert 2 μg of RNA to cDNA in a total reaction volume of 20 μL.

### 2.8. Quantitative RT-qPCR

Quantitative RT-qPCR was performed using SYBR Green PCR Master Mix (Bio-Rad, Bio-Rad Laboratories, Inc., Hercules, CA, USA) on a Bio-Rad Prime PCR instrument. The qRT-PCR conditions were 40 cycles of 30 s at 95 °C, 15 s at 60 °C, and 15 s at 72 °C. The primers used in these studies are listed in Appendix A.

### 2.9. Immunostaining

For staining, the cell cultures were washed twice in 1× phosphate-buffered saline (PBS, pH 7.2) (Welgene, Inc., Gyeongsan, Korea) and fixed with 4% Paraformaldehyde in PBS for 15 min at room temperature. The cells were washed three times with 1× PBS and permeabilized with 0.25% Triton-X-100 (USB Corporation, Thermo Fisher Scientific, Inc., Waltham, MA, USA) in PBS for 10 min at room temperature. Prior to blocking, the cells were washed three times with 1× PBS. Then, the cells were blocked with blocking solution containing 1% Bovine Serum Albumin (Amresco, Dallas, TX, USA), 22.52 mg/mL of glycine (Affymetrix, Thermo Fisher Scientific, Inc., Waltham, MA, USA), and 0.1% Tween 20 (Affymetrix, Thermo Fisher Scientific, Inc., Waltham, MA, USA) or 5% normal goat serum (Thermo Fisher, Waltham, MA, USA) in PBS for 60 min. Subsequently, the cells were stained with the appropriate primary antibodies diluted in blocking solution overnight at 4 °C. After primary antibody incubation, the cells were washed in PBS with 0.1% Tween-20 (PBST) three times and incubated with secondary antibodies for 2 h at room temperature. The cells were then washed with PBST three times and incubated with 1 μg/mL of DAPI (Sigma-Aldrich, Inc., Saint Louis, MO, USA) for 5 min at room temperature to stain the nuclei. The samples were visualized using fluorescence microscopy (IX71S1F3, Olympus, Tokyo, Japan). All the antibodies used in this study are listed in Appendix A.

### 2.10. Calcium Signaling

Calcium imaging was performed on mouse astrocyte C8-D1a, which induced neuron-like cells after 15 days of neuronal induction using Rhoda-2-AM (Sigma-Aldrich, Inc., Saint Louis, MO, USA) at a concentration of 2 μM. Imaging was performed in Live-Imaging Solution (Invitrogen, Thermo Fisher Scientific, Inc., Waltham, MA, USA), and calcium signaling was evoked using 10 mM KCL. Images were acquired at 30 frames/s using a scientific CMOS camera. The microscope was controlled by Micro-Manager software and the image processor ImageJ. The samples were visualized using fluorescence microscopy (IX71S1F3, Olympus, Tokyo, Japan). Changes in fluorescence were measured for individual cells, and the average of the first 10 time-lapse images for each region of interest (ROI) was defined as the initial fluorescence (F_0_), as shown in Appendix A.

### 2.11. Generation of Reactive Astrocytes

For the generation of reactive astrocytes, primary human cortical astrocytes were cultured in Neurobasal Plus media in the presence of 10% FBS, 1% N2, and 100 ng/mL of Liposaccharides (LPS) from *Escherichia coli* K-235 (Sigma-Aldrich, Inc., Saint Louis, MO, USA) for 18 h.

## 3. Results

### 3.1. Cell Characterization of Mouse Astrocyte C8-D1a

Because astrocytes show varying degrees of complexity in the brain based not just on their morphology, but also their spatial and temporal locations and origin [51,52,53], we selected a homogeneous mouse astrocyte C8-D1a cell line for our study. The mouse astrocyte C8-D1a was previously described as being isolated from the cerebella cortex of an eight-day post-natal brain and was characterized as containing both type 1 and type 2 astrocytes based on cell shape [54]. Although mouse astrocyte C8-D1a spontaneously transformed into an immortalized cell line, they have been routinely used to study both the physiological functioning of astrocytes and their pathological role in neurodegenerative disease [55,56].

To reconfirm the previously reported cell characterization, we carried out an initial cell characterization using immunostaining. Our in-house characterizations revealed that the mouse astrocyte C8-D1a was mainly GFAP positive, with more than 98% showing positive immunostaining (Appendix A). Less than 4% of the mouse astrocyte C8-D1a stained positive for neuronal markers, such as Tuj1, DCX, or MAP2 (Appendix A). Our in-house immunostaining characterizations also revealed that the mouse astrocyte C8-D1a cell line was a highly pure culture without significant contamination of neuronal cells (Appendix A). To rule out contamination from neuronal cells, we compared nestin marker expression with that in NSE-34, a murine motor neuronal progenitor cell line. A few cells showed a low level of scattered staining for the stem cell marker nestin (Appendix A), which we ruled out as background staining, due to its much lower intensity compared with that of the NSE-34 neuronal stem cells. Additionally, we observed that the cultured astrocytes formed a compact layer of cells over time, which is typical of mouse astrocyte morphology.

### 3.2. Screening of Small-Molecule Chemical Library Reveals Ruxolitinib as a Potential Inducer of Astrocyte-to-Neuronal Reprogramming

The results from previous protocols for converting astrocytes to neurons suggested that modulating the pathways associated with the neuron glial switch would be useful. Small molecules from the chemical library were screened over a basal cocktail of forskolin, CHIR99021, RepSox, and LDN193189, which were chosen due to their roles in astrocyte-to-neuron reprogramming in previously published protocols [4,42,43] (Appendix A). Each of the 93 small molecules in the library was screened at a concentration of 5 μM for four days. Our initial assessment of each molecule’s ability to induce neuronal reprogramming was based on the appearance of a neuronal morphology. We corroborated those assessments by immunostaining for the neuron-specific Tuj1 marker.

Our screening study revealed three important findings. First, the basal cocktail alone did not effectively generate neurons in four days. Cells containing only the basic cocktail took on an elongated morphology that resembled that of fibrous astrocytes (Appendix A) [57]. Second, DAPT (Appendix A) and Y26732 (Appendix A) effectively induced neurogenesis from mouse astrocytes when they were combined with the basal cocktail, and both of them had previously been reported to do so in primary astrocyte cells [4,16,42]. Third, in combination with the basal cocktail, ruxolitinib, a 1/2 JAK/STAT inhibitor, generated a cluster of rounded Tuj1^+^ cells from mouse astrocytes, which we are the first to report (Appendix A). Thus, we obtained a pool of seven small molecules that we tested in various combinations to find an efficient cocktail for inducing neuronal reprogramming in the mouse astrocytes.

### 3.3. Small Molecule Cocktail Induces Neuronal Programming in Mouse Astrocytes in Four Days

During optimization, we found that combining all seven chemicals produced an elongated morphology [57]. Through the process of elimination, we found that removing forskolin prevented the elongated morphology. Additionally, removal of LDN193189 from the complete seven-chemical cocktail reduced the compacted morphology (Appendix A). Based on those results, we used a five-chemical cocktail (5C), viz. CHIR99021, RepSox, Y26732, DAPT, and ruxolitinib, to generate neurons more efficiently than the other combinations (Appendix A). To further enhance that efficiency, we investigated the possibility of activating SHH signaling, which has been reported to improve the efficiency of neural reprogramming [58,59]. Indeed, we found that adding SAG, an SHH signaling activator, to the 5C cocktail increased the number of Tuj1^+^ cells (Appendix A). We used these findings and constituted our NIM of six chemicals (6C), SAG, CHIR99021, RepSox, Y26732, DAPT, and ruxolitinib, along with the additional growth factors BDNF, GDNF, and NT3 (Figure 1a,b and Appendix A). We further found that the use of a Neurobasal Plus medium improved the conversion efficiency compared to that of DMEM: F12 and N2.

To evaluate the reprogramming efficacy of the NIM after four days of treatment, we stained the cells with the early neuronal markers Tuj1 and DCX. Those staining results indicated that 82 ± 6% of the cells were positive for Tuj1 expression (Figure 1b). The neuronal purity of differentiated cells on day four, based on Tuj1 and DCX co-expression, was 94 ± 3% (Figure 1c). In addition to early neuronal markers, further characterization also showed the expression of pan-neuronal markers MAP2 and NeuN (Figure 1e and Appendix A). Evident by neuronal morphology and the expression of four key markers, we define these cells as neuron-like cells differentiated for four days (NLC4). In the same period (four days), control mouse astrocytes maintained without any small-molecule treatment and 1% dimethyl sulfoxide showed no morphological changes. These results suggest that our NIM containing six small molecules and growth factors successfully induced neuronal programming in mouse astrocytes.

### 3.4. Maturation of Reprogrammed Cells to Generate Neuron-like Cells

During the four-day neuronal induction period, the mouse astrocytes gradually took a neuron-like morphology, but they lacked connections between neurites, which is the hallmark of matured, functional neurons (Figure 1c and Appendix A). Since NIM beyond day four did not further improve neural connectivity, we changed the NIM to neuronal maturation medium (NMM) which would ensure the functional maturation and survivability of NLC4. For the constitution of NMM, we tried each of the molecules from our pool to find molecules that could improve the maturation process. We found that Y27632, CHIR99021, and SAG in combination with growth factors (dbcAMP, BDNF, NT3, GDNF, and B27) in Neurobasal Plus medium generated cells with matured phenotype showing better neurite growth and connectivity, along with the expression of four pan-neuronal markers at day ten (Figure 1d,e). We define these neuron-like cells differentiated for ten days as NLC10. The RT-qPCR analysis revealed an increase in key neuronal transcription factors (NgN2, Ascl1, Tbr1, and NeuroD1) associated with neuronal reprogramming (Figure 1f). Furthermore, the RT-qPCR analysis for the neuronal markers NeuN, MAP2, and Syn1 at day four and day ten showed significant expression, as compared to the control (Figure 1g). DCX expression was undetectable on day ten, but we observed an increase in Syn1 expression, suggesting that the neuronal networks were successfully connected. Our immunostaining data corroborate the RT-qPCR data, showing the expression of the key neuronal markers Tuj1, NeuN, and MAP2. At day ten, we still detected DCX expression on the protein level, which did not match the RT-qPCR data (Figure 1e,g). Together, our data demonstrate that the maturation medium sustained the expression of the neuronal transcription factors required for neural maturation, and we obtained neuron-like cells from astrocytes.

### 3.5. Role of Small Molecules in Neuronal Conversion

To investigate the role of each of the small molecules in our 6C cocktail, we analyzed the effects of removing them individually from the main chemical cocktail. The initial assessment demonstrated that removing any one of the small molecules decreased the number of newly reprogrammed neurons only marginally, with a decrease in efficiency of 5–20%, as reflected by the number of Tuj1^+^ cells. We found that when either RepSox or DAPT was excluded from the main cocktail, there was a delay in generating reprogrammed neurons (Figure 2a).

Consequently, excluding both RepSox and DAPT had the most detrimental effect on neuronal reprogramming of our cocktail, with a decrease in conversion efficiency of 15–20% over four days (Figure 2b,c). On the other hand, removal of ruxolitinib had only a minor effect on the overall efficiency of the conversion (Figure 2b,c). The exclusion of the remaining three chemicals, SAG, CHIR99021, and Y26732, caused a significant drop in efficiency that was not as significant as that for RepSox and DAPT. Thus, DAPT and RepSox are key chemicals in our cocktail for converting mouse astrocytes into neurons.

### 3.6. Functional Characterization of Astrocyte-Derived Neuron-like Cells

To assess the neuronal subtype and functionality of NLC10 generated using NMM, we performed a RT-qPCR analysis for several neuronal subtype markers at day four and day ten. The persistent expression of vGLUT1 throughout the reprogramming process suggested the possible fate of these cells to be glutaminergic (Figure 3a). Additionally, we observed a punchated expression of matured neuronal marker Syn1 at day 12 indicating the functional maturation of the neuron-like cells (Figure 3b). Furthermore, a dual immunostaining analysis of neurotransmitter markers and MAP2 indicated that more than 98% of the neuron-like cells were both MAP2 and vGLUT1 positive (Figure 3c,d). However, we did not detect the expression of other neurotransmitters, such as TH and GAD67, along with MAP2 (Figure 3d), which further highlights the glutaminergic fate of neuron-like cells.

To evaluate whether the neuron-like cells generated from mouse astrocyte C8-D1a could elicit action potential, we investigated KCl-evoked calcium signaling. After 15 days of reprogramming, a topical application of 10 mM KCl evoked calcium flux in about 38% of the cells (Figure 4a,b). To compare the functional activity of NLC15s reprogrammed from mouse astrocyte C8-D1a and control cells, the calcium signals of at least 10 randomly chosen NLC15s were analyzed, and their maximal fluorescence intensity was plotted (Figure 4c). We observed three unique profiles of depolarization (Figure 4d). In group 1, we found a sharp rise in calcium influx followed by the depolarization of neurons after 10 to 15 s. In group 2, we found a more gradual rise in calcium intake followed by a slower depolarization.

In group 3, we found that calcium instantly taken up by NLC15s was gradually released in the same pattern as the calcium influx. Altogether, these finding suggest that the reprogrammed cells showed an average peak (Δ*F_max_*/*F*_0_) of 1.25 ± 0.6 (Figure 4e). Thus, the mouse astrocytes C8-D1a-derived NLCs showed KCl-induced depolarization with different sensitivity for the action potential.

## 4. Discussion

Astrocytes are an attractive target for in vivo reprogramming for the treatment of neurodegenerative diseases, due to their spatial proximity and common developmental lineage to neurons [24]. By modulating key pathways associated with the neuro-glial switch, we aimed to generate a rapid protocol for converting astrocytes into neurons. To that end, we used the mouse astrocyte C8-D1a cell line. Although not without its limitations, the mouse astrocyte C8-D1a provides a homogeneous population of cells that can be used as a platform to investigate small molecules for their potential to convert astrocytes into neurons. From a reprogramming point of view, it is advantageous to use cell lines, because the existing reprogramming protocols suffer from inconsistency and a low conversion ratio that possibly arises from the heterogeneity of the primary cells [60]. For the protocol proposed here, we used a cell line to better assess the functional role of the small molecules we tested because cell lines always give rise to a homogeneous cell population. Additionally, in the context of cellular reprogramming, the highly reproducible results generated herein form a basis for future exploratory experiments with primary astrocytes under both normal and diseased physiological conditions. Our protocol results in nearly 83% cell reprogramming efficiency, which is on par with other protocols (Table 1).

Numerous reprogramming protocols, using primary mouse or human astrocytes as a source to generate neurons, make use of either HDAC inhibitors or BET inhibitors (or both) to promote neurogenesis [4,42,43,44]. Although the use of HDAC inhibitors greatly improves the efficiency of astrocyte-to-neuron conversion, prolonged exposure to epigenetic modifiers causes cell toxicity. To prevent toxicity, we designed a chemical cocktail devoid of epigenetic modifiers that can induce the reprogramming of astrocytes to neurons. Our results demonstrate that the use of an HDAC inhibitor, such as VPA, is unnecessary for neuronal conversion of the mouse astrocyte C8-D1a cell line. A similar approach was reported by L. Gao et al. using primary human astrocytes [16], but we are the first to demonstrate that a cell line, in this case, mouse astrocytes, can be efficiently converted into neurons without the use of any epigenetic modifier.

To develop a rapid protocol for efficiently generating neuron-like cells from mouse astrocytes without using an epigenetic modifier, we initially conducted a screening assay using a basal cocktail of forskolin, CHIR99021, RepSox, and LDN193189. We also found that ruxolitinib, a JAK-STAT inhibitor, in combination with FCRL could effectively convert mouse astrocytes into neuron-like cells, as shown by the appearance of clusters of rounded Tuj1^+^ cells. The JAK-STAT pathway has been well documented to have a key role in differentiation and maintenance of astrocytic fate and regulation [61,62]. The introduction of the SHH signaling activator SAG further enhanced the efficiency, and the resulting 6C chemical cocktail rapidly and efficiently induced the reprogramming of mouse astrocytes into NLCs in just four days. Studies of RNA sequences by Nig et al. suggest that the JAK/STAT pathway can be effectively downregulated without the need for a specific JAK/STAT inhibitor, indicating the chemical redundancy of ruxolitinib [63]. However, we saw a small but significant increase in the neuronal efficiency conversion of mouse astrocytes with inclusion of ruxolitinib. To our knowledge, ruxolitinib has not been used for the conversion of astrocyte to neurons till date [Table 1]. Additionally, SAG was a key part of the maturation and maintenance of NLCs from mouse astrocytes. WNT signaling induced by the addition of CHIR99021 forms the basis of all known astrocyte-to-neuronal reprogramming at both the induction and maturation stages [Table 1]. Two of the 6C small molecules, DAPT and RepSox, played a major role in the neuronal conversion of mouse astrocytes and have been reported in multiple protocols using different chemical combinations, but not together. We hypothesize that combining those two molecules with CHIR99021 produced a rapid and efficient neural conversion process, without the need for an epigenetic modifier. Thus, our protocol, in addition to two previous human primary astrocyte-to-neuron protocols [Table 1], suggests that those three molecules could form the core of a single-step protocol for the neuronal reprogramming of both mouse and human astrocytes to neurons.

Finally, in line with previously existing astrocyte-to-neuron conversion protocols, mouse astrocyte C8-D1a were largely converted to a pure culture of neuron-like cells with the properties of glutaminergic neurons [4,16,42,43]. To assess whether we could generate another subtype after neural induction, we treated NLC4 with known dopaminergic factors, such as SSH signaling activators SAG, fgf8b, TGF-b3, and vitamin C. Even though those factors are well known to induce a dopaminergic fate, we could not find any dopaminergic cells, indicating that the cells were primed for a glutaminergic fate after induction for four days (data not shown). Furthermore, we attempted to generate neuron-like cells from human primary cortical astrocytes using our neural induction media. However, the reprogramming of human astrocytes into neurons was less reproducible and its conversion efficiency was low (Appendix A). Upon further investigation, we found that human primary cortical astrocytes were highly heterogeneous in terms of the GFAP and S100b expression (Appendix A). Therefore, we propose that an astrocyte subtype with an expression pattern of GFAP and S100b, similar to that in the mouse astrocyte C8-D1a, would be more receptive than other astrocytes to our cocktail and reprogramming protocol. Nevertheless, in addition to the above results, we assessed the reprogramming potential of our chemical cocktail under LPS-mediated astrocyte reactivity, which resembles that of the human reactive astrocytes present in Parkinson’s disease’s pathophysiology [64,65,66]. The generation of reactive astrocytes using LPS was confirmed using enhanced GFAP expression [67] (Appendix A) and the application of our 6c chemical for four days on human reactive astrocytes generated neurons similar to NLC4 (Appendix A). Along with reactive human astrocytes, there is also a need to investigate the effect of our cocktail on other glial cells present in the central nervous system. However, since the reprogramming potential of a cocktail depends on the starting cell type and needs to be optimized accordingly, we hypothesize that our cocktail may not have a similar effect to the one as seen on astrocytes. Although there are very few reported protocols on the conversion of cells from ectodermal lineages to neurons, we find that chemical cocktails used for each conversion varies, depending on the cellular identity [68,69] (Appendix A).

Taken together, our results demonstrate that our 6C cocktail followed by maturation using three small molecules could generate a pure culture of neuron-like cells with functional glutaminergic neuronal property from mouse astrocyte C8-D1a. Our chemical cocktail was obtained through in-house screening on a homogeneous mouse astrocyte cell line and can serve as a foundation for further investigations of human astrocyte-to-neuron conversion. Moreover, our results from the use of mouse astrocyte C8-D1a highlights the importance of using a homogeneous cell culture for rapid, pure, and highly efficient cellular conversion. In this regard, our future studies will look to introduce additional pre-treatment steps prior to cellular reprogramming, in order to obtain a homogenous starting cell culture.

## Figures and Tables

**Figure 1 biomedicines-10-00928-f001:**
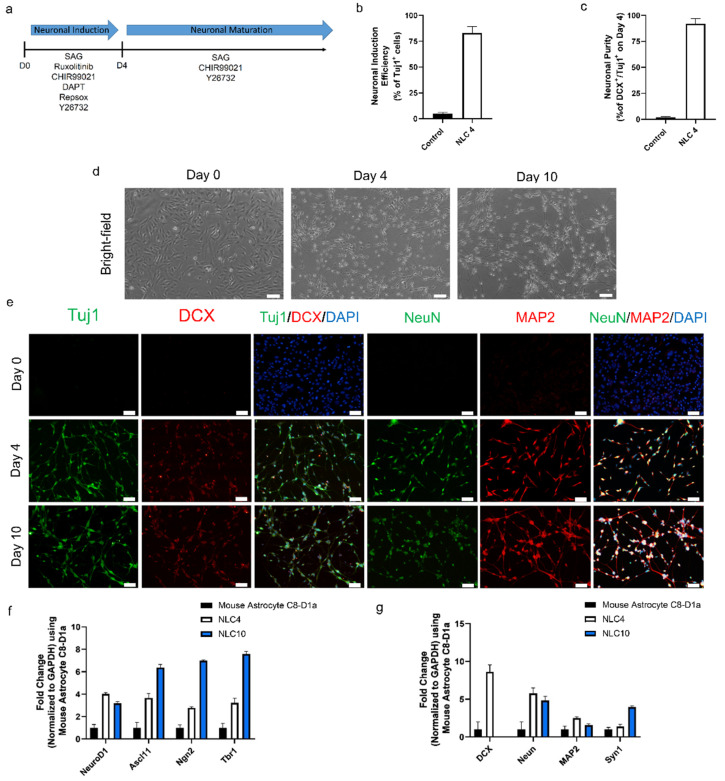
(**a**) Schematic design of the protocol for converting mouse astrocytes into neuron-like cells. (**b**) Neuronal conversion efficiency determined on day 4 after differentiation for NLC4 normalized to control astrocytes. (**c**) Neuronal purity determined on day 4 after differentiation for NLC4 normalized to control astrocytes. (**d**) Bright-field images of astrocytes–converted neuron-like cells on days 0, 4, and 10. Scale bar, 100 μm. (**e**) Immunofluorescence labeling with Tuj1, Map2, NeuN, and DCX on astrocytes–converted neuron-like cells on days 0, 4, and 10. Scale bar, 50 μm. (**f**) mRNA transcript levels of key neuronal transcription factors were assessed by qRT–PCR on days 0, 4, and 10. (**g**) mRNA transcript levels of key neuronal markers were assessed by qRT–PCR on days 0, 4, and 10.

**Figure 2 biomedicines-10-00928-f002:**
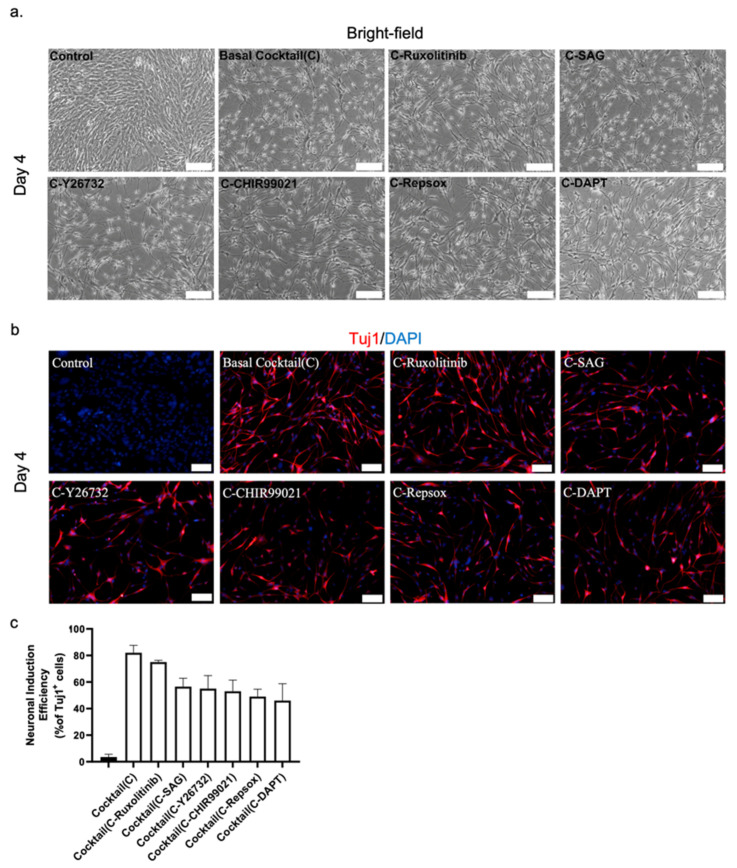
(**a**) Representative bright-field image of NLCs from mouse astrocytes. The role of the small molecules was investigated by individually removing them from the 6C cocktail. Scale bar, 100 μm. (**b**) Immunostaining with Tuj1 to investigate the role of the small molecules. Scale bar, 50 μm. (**c**) Neuronal conversion efficiency.

**Figure 3 biomedicines-10-00928-f003:**
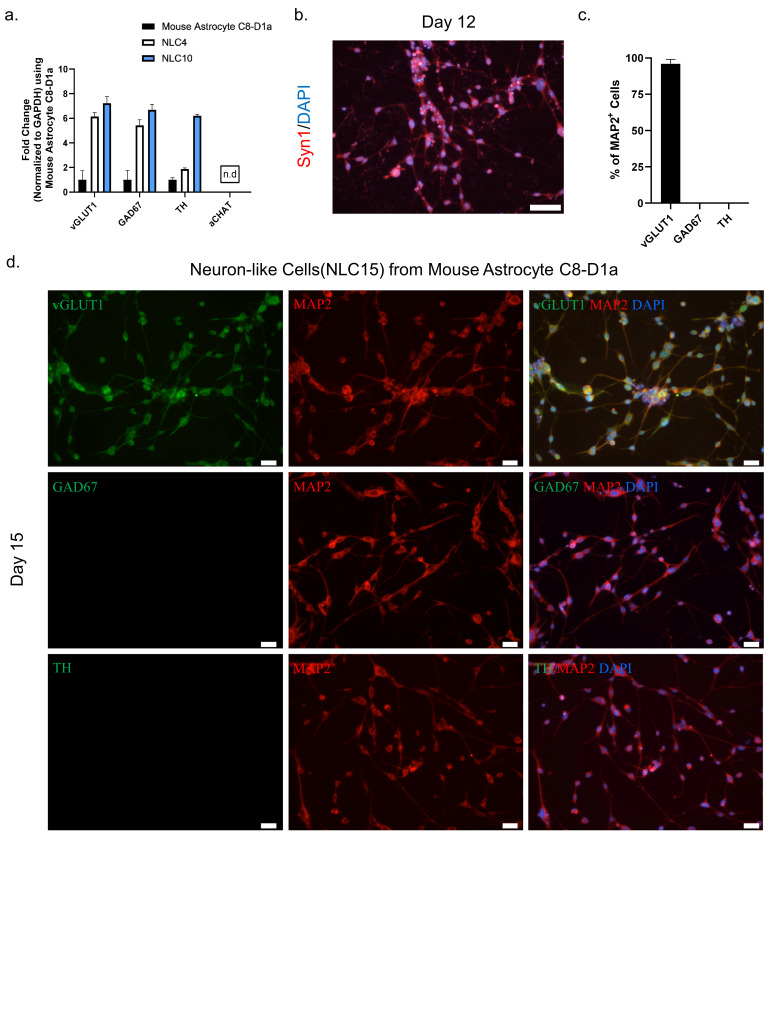
(**a**) mRNA transcript levels of the neurotransmitters vGLUT1, GABA, TH, and aCHAT were assessed by qRT–PCR on days 0, 4, and 10. (**b**) Immunostaining labeling of neuron-like cells with the marker Syn1 on day 12 Scale bar, 50 μm. (**c**) Quantification of neuronal subtypes. (**d**) Immunostaining labeling of neurons-like cells with the neuronal markers MAP2, vGLUT1, GABA, and TH on day 15. Scale bar, 20 μm.

**Figure 4 biomedicines-10-00928-f004:**
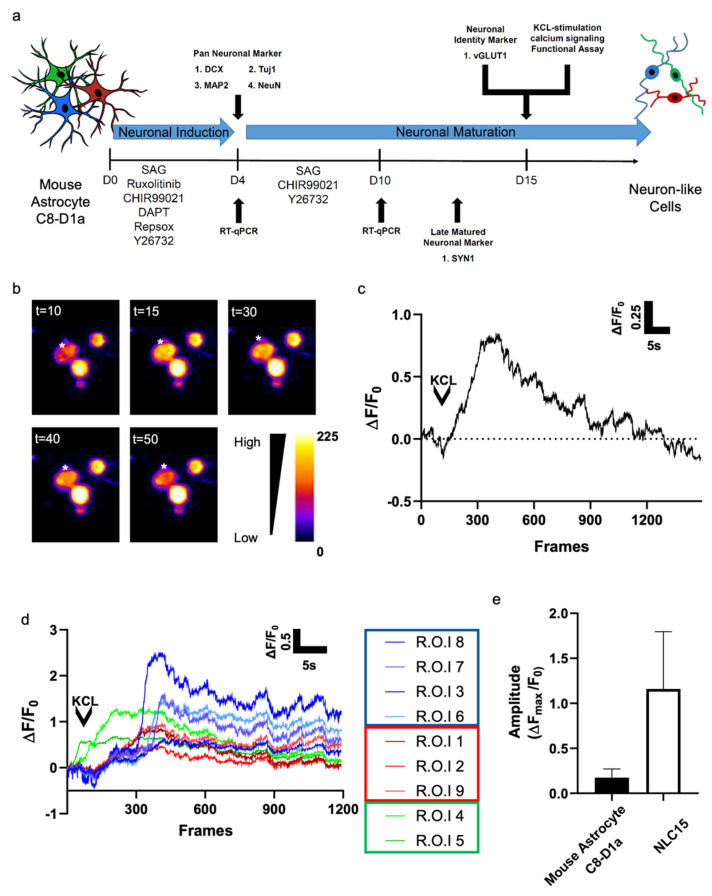
(**a**) Schematic illustration of the experimental timeline for neuronal reprogramming and functional assessment. (**b**) Fluorescent image of a cell loaded with 2 μM Rhoda 2-AM over different time intervals. Calcium flux was stimulated using 10 mM KCL. The asterisk indicates the regions in which the fluorescence measurements were performed. Selected pseudo-color frames are baseline-subtracted images (ΔF) of the cell shown. (**c**) Increases in calcium, plotted as ΔF/F0, measured over the regions indicated in a are shown over time (30 frames/s). (**d**) Increases in calcium, plotted as ΔF/F_0_, measured using multiple ROIs shown over time (30 frames/s). (**e**) The average Δ*Fmax*/*F_0_* from (**c**) was determined and compared with that from (**d**).

**Table 1 biomedicines-10-00928-t001:** List of astrocyte to neuron conversion protocols.

Sr No.	Cell Origin	Small Molecule	Neuronal Induction Duration	Neuronal Conversion Efficiency * (Purity) **	Characterization	Reference
**1**	Human primary astrocyte	Valpoic Acid CHIR99021 Repsox iBET151 ISX-9 Forskolin	18 days	8 % MAP2a^+^ (70% MAP2a^+^)	Tuj1^+^, DCX^+^, Tau^+^, NeuN^+^, GABA^+^, vGLUT1^+^, MAP2a^+^	[42]
**2**	Human primary astrocyte	CHIR99021 Valpoic Acid DAPT LDN193189 SB431542 TTNPB Thaizovin SAG Puroamphamine	8 days	67% Tuj1^+^	Tuj1^+^, DCX^+^, NeuN^+^, Syn1^+^, GABA^+^, vGLUT1^+^, MAP2^+^	[4]
**3**	Mouse primary astrocyte	Valpoic Acid CHIR99021 Repsox	14 days	24% NeuN^+^	Tuj1^+^, DCX^+^, NeuN^+^	[43]
**4**	Mouse Primary Astrocyte	Forskolin ISX9 CHIR99021 iBET151	16 days	89.2 ± 1.4 % Tuj1^+^	Tuj1^+^, DCX+, Tau+, NeuN+, GABA+, vGLUT1+, MAP2a+	[44]
**5**	Mouse Astrocyte C8-D1a cell line	SAG CHIR99021 DAPT Ruxolitinib RepSox Y26732	4 days	82 ± 6% Tuj1^+^ (94 ± 3% Tuj1^+^/DCX^+^)	Tuj1^+^, DCX^+^, MAP2^+^, NeuN^+^ (vGLUT1^+^, GAD67^+^, TH^+^) ***	This study

* Conversion efficiency is defined as the ratio of marker positive cells to the initial number of seeded cells in each visual field. ** Purity is defined as the percentage of marker-induced cells in total cells stained by DAPI. *** Markers in parenthesis were characterized by qRT-PCR.

## Data Availability

Data regarding this study are available on request to the corresponding author.

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
