# Peer review of "Generation of a Pure Culture of Neuron-like Cells with a Glutamatergic Phenotype from Mouse Astrocytes"

_biomedicines, 2022, doi:10.3390/biomedicines10040928_

Round 1
Reviewer 1 Report
Dear Author,
the manuscript titled "Generation of a pure culture of neuron-like cells with a glutamatergic phenotype from mouse astrocytes" is very interesting and well written.
I have just few comments in order to improve the quality of your paper.
1- In the introduction section, more details concerning previous study both on astrocytes or glial cells should be quoted. For example: 10.1038/ncb2843; 10.1016/j.stem.2011.11.015; 10.1016/j. stemcr.2014.10.007.
2- Figure 1, panel D: I suggest you to split the red and green channel in order to better show the differences.
3- Please, use italics for latin words, such as "in vivo" and "in situ" (page 2, line 46).
Author Response
Point-by-point Response to the Reviewers’ Comments
Reviewer 1:
Comments to the authors:
The manuscript titled "Generation of a pure culture of neuron-like cells with a glutamatergic phenotype from mouse astrocytes" is very interesting and well written.
I have just few comments in order to improve the quality of your paper.
- In the introduction section, more details concerning previous study both on astrocytes or glial cells should be quoted. For example: 10.1038/ncb2843; 10.1016/j.stem.2011.11.015; 10.1016/j. stemcr.2014.10.007.
Answer:
We acknowledge the reviewer’s concern on including the details of previously reported studies in the introduction section. We have taken the suggestion and dedicated a paragraph pertaining to reprogramming studies in Astrocyte and Glial.
(Page-2, lines 48-59, red font).
“In this regard, initial principal studies have successfully demonstrated the neuronal conversion of both human and mouse astrocyte to neurons or neuroblasts in vitro [13-16]. Employing in vivo ectopic expression of pluripotent transcription factor such as Sox2 on astrocytes, successfully generates a pool of progenitor cells which then undergo neurogenesis [17-20]. On the other hand, the more attractive and preferable approach is the direct reprogramming using ectopic expression of neuronal transcription factor, NeuroD1, which sufficiently generates neurons without the appearance of a progenitor like state [21-23]. Interestingly, employing similar investigative strategies on resident glial cells have yield similar neurogenic potential as that of astrocytes [24-26]. In summary, in vivo reprogramming of glial and astroglial cells appears to be a promising strategy in regenerative medicine, opening avenues for further optimization under neurodegenerative physiological condition.”
- Figure 1, panel D: I suggest you to split the red and green channel in order to better show the differences.
Answer:
We acknowledge the reviewer’s concern regarding splitting the red and green channel in Figure 1, panel d. For a better representation of the figure, we have split panel d into two parts. As a result, previously labelled figure 1d is now figure 1d and 1e, while figures 1e & 1f are now figure 1f and 1g respectively.
- Please, use italics for latin words, such as "in vivo" and "in situ" (page 2, line 46).
Answer:
We acknowledge the reviewer’s concern regarding the use of italics for Latin words. We have made changes wherever we have used the “in vivo” and “in vitro”.
Reviewer 2 Report
In this paper Fernandes et al has demonstrated a faster and more accurate reprogramming method using a small molecule cocktail for conversion of astrocytes to the neuron like cells. In the present study authors used a secondary astrocyte mouse cell line, C8-D1a, to avoid the heterogeneity of the primary astrocyte culture or contamination of other cell types. The C8-D1a cell line was converted to Tuj1 positive cells within 4 days of treatment using a cocktail of small molecules. Authors have demonstrated that there was no significant pre contamination of progenitor cells and adult neurons in the C8-D1a culture. Therefore, authors claimed that the cocktail of small molecule might have therapeutic potential for the treatment of degenerative diseases, where neurons are progressively lost in the brain. Neuro progenitor cells are confined in the certain regions of the brain. Therefore, targeting the astrocytes, mostly abundant cells in the brain, to convert to functional neurons must be a good approach for next generation treatment of neurodegenerative diseases. But there are number of concerns to be addressed in this paper before publication.
According to the Ben Barres’s group the astrocyte cell represents 30% of the total cells in the central nervous system. In the degenerative brain, microglia engulfs dead neurons and gets activated. Activated microglia induces several inflammatory pathways and secretes several inflammatory cytokines and complements components, which convert the resting astrocyte to the activated disease associated form (A1 type). This group showed that activated astrocytes are mostly abundant in the degenerative brain, which loses its homeostatic form and causes progressive neurodegeneration. Therefore, the question is whether activated form of the astrocyte will behave the same way as resting/homeostatic astrocyte does, when treated with the small molecule cocktail? Authors need to provide experimental evidences.
How does this cocktail of small molecules/ chemicals affect other cell types like microglia? Did authors check it in the glial cell, because brain contains both astrocyte and glia cells?
How much these small molecules penetrate the brain/ cross blood brain barrier were not discussed in this manuscript.
Author Response
Comments to the authors:
In this paper Fernandes et al has demonstrated a faster and more accurate reprogramming method using a small molecule cocktail for conversion of astrocytes to the neuron like cells. In the present study authors used a secondary astrocyte mouse cell line, C8-D1a, to avoid the heterogeneity of the primary astrocyte culture or contamination of other cell types. The C8-D1a cell line was converted to Tuj1 positive cells within 4 days of treatment using a cocktail of small molecules. Authors have demonstrated that there was no significant pre contamination of progenitor cells and adult neurons in the C8-D1a culture. Therefore, authors claimed that the cocktail of small molecule might have therapeutic potential for the treatment of degenerative diseases, where neurons are progressively lost in the brain. Neuro progenitor cells are confined in the certain regions of the brain. Therefore, targeting the astrocytes, mostly abundant cells in the brain, to convert to functional neurons must be a good approach for next generation treatment of neurodegenerative diseases. But there are number of concerns to be addressed in this paper before publication.
According to the Ben Barres’s group the astrocyte cell represents 30% of the total cells in the central nervous system. In the degenerative brain, microglia engulfs dead neurons and gets activated. Activated microglia induces several inflammatory pathways and secretes several inflammatory cytokines and complements components, which convert the resting astrocyte to the activated disease associated form (A1 type). This group showed that activated astrocytes are mostly abundant in the degenerative brain, which loses its homeostatic form and causes progressive neurodegeneration.
- Therefore, the question is whether activated form of the astrocyte will behave the same way as resting/homeostatic astrocyte does, when treated with the small molecule cocktail? Authors need to provide experimental evidences.
Answer
We acknowledge the reviewer’s concern regarding the reprogramming potential of our chemical cocktail on reactive astrocytes. To address the question, we generated reactive astrocytes by treating primary human cortical astrocytes with 100 ng/ml of Lipopolysaccharides (LPS) from Escherichia coli K-235 for 18 hrs. LPS-induced reactivity of astrocytes was confirmed by the enhanced expression of astrocyte marker GFAP (Supplementary figure S5b) which appears to be in line with previously reported studies (Brahmachari S, Fung YK, Pahan K. Induction of glial fibrillary acidic protein expression in astrocytes by nitric oxide. J Neurosci. 2006;26(18):4930-4939. doi:10.1523/JNEUROSCI.5480-05.2006). The method to generate reactive astrocytes has been mentioned in the materials and method section 2.11.
(Page-13, lines 436-442, red font).
“2.11. Generation of reactive astrocytes
For the generation of reactive astrocytes, primary human cortical astrocytes were cultured in Neurobasal Plus media in presence of 10% FBS, 1% N2 and 100 ng/ml of Liposaccharides (LPS) from Escherichia coli K-235 (Sigma-Aldrich) for 18 hrs.”
After LPS-mediated activation, astrocytes were treated with our chemical cocktail for a period of four days for neural induction. The treatment generated neuron like cells (NLCs) which looks similar to NLC4 generated without LPS treatment in terms of both morphology and Tuj1/MAP2 marker expression (Supplementary figure S5c & S5d). This study is a qualitative analysis based on two sets of experiments and further investigations are required to analyze the quantitative and long-term effect of our chemical treatment on LPS-induced reactive astrocytes. We have added these results and observation in the discussion section of the manuscript.
(Page-13, lines 436-442, red font).
“Nevertheless, in addition to the above results, we assessed the reprogramming potential of our chemical cocktail under LPS-mediated astrocyte reactivity which resembles that of reactive astrocytes present in Parkinson’s diseases pathophysiology [64-66]. The generation of reactive astrocytes using LPS was confirmed using enhanced GFAP expression [67] (Supplementary Figure S5b) and the application of our 6c chemical for four days on reactive astrocytes generated neurons similar to NLC4 (Supplementary Figure S5c & d).”
All the results from this study have been included in a new Supplementary Figure S5.
Supplementary Figure S5 a) Schematic design for assessing neuronal reprogramming potential of our chemical cocktail on LPS-induced human cortical reactive astrocytes. b) Immunostaining of chemically induced astrocytes with astrocyte marker GFAP pretreated with or without LPS. Scale bar, 50 μm. c) Representative bright-field images of chemically induced astrocytes over four days of exposure to our neuronal induction cocktail pretreated with or without LPS. Scale bar, 100 μm. d) Immunofluorescence labeling chemically induced astrocytes with Tuj1and MAP2 after four days of exposure in neuronal induction cocktail pretreated with or without LPS. Scale bar, 50 μm.
Similar to our results, there are previous studies which support the ability of reactive astrocytes to generate neurons in presence small molecules or transcription factors like NeuroD1. Interestingly, RNA-seq data from small molecules mediated reprogramming of astrocytes shows a marked rise in astrocyte-associated genes during the initial course of reprogramming which then subside over the rest of the course of reprogramming (Gao L, Guan W, Wang M, Wang H, Yu J, Liu Q, Qiu B, Yu Y, Ping Y, Bian X, Shen L, Pei G. Direct Generation of Human Neuronal Cells from Adult Astrocytes by Small Molecules. Stem Cell Reports. 2017 Mar 14;8(3):538-547. doi: 10.1016/j.stemcr.2017.01.014. Epub 2017 Feb 16. PMID: 28216149; PMCID: PMC5355633.). This evidence has generated a school of thought that astrocyte reprogramming from homeostatic/resting astrocyte goes through an intermediate stage of reactive astrocytes before acquiring neural transcriptomic signature.
- How does this cocktail of small molecules/ chemicals affect other cell types like microglia? Did authors check it in the glial cell, because brain contains both astrocyte and glia cells?
Answer
We acknowledge the reviewer’s concern, regarding testing our cocktail on other cell types found in the brain such as microglia, Schwann cells etc. We haven’t checked the effect of this chemical cocktail on other cells present in the brain. Since the reprogramming ability of the chemical cocktail depends on the initial cell type and must be optimized accordingly, it is expected that the other glial cells cannot be converted to neuron when the same protocol is applied. For example, the conversion condition of muller glia to neuron is known to contains Forskolin, ISX9, CHIR99021, I-BET151, and Y-27632, which is quite different from the current condition. We have included a table below showing the chemical cocktails required for generating neurons from different ectodermal linage cells, thus highlighting the significance of initial cell state. Furthermore, to investigate the effect of our cocktail on other cell types present in brain, we plan to use both an in vitro tri-culture system and a mice model to check the applicability of this cocktail for in vivo reprogramming. However, the main purpose of this study is to highlight the importance of culture homogeneity for a rapid and efficient reprogramming and develop a platform which would help in the better functional assessment of molecules to reprogram astrocytes to neurons. Accordingly, we have addressed this concern in the discussion section of the manuscript.
(Page-13, lines 442-449, red font).
“Along with reactive human astrocytes, there is also a need to investigate the effect of our cocktail on other glial cells present in the central nervous system. However, since the reprogramming potential of a cocktail depends on the starting cell type and needs to be optimized accordingly, we hypothesis that our cocktail may not have a similar effect as seen on astrocytes. Although there are very few reported protocols on the conversion of cells from ectodermal lineages to neurons, we find that chemical cocktails used for each conversion varies depending on the cellular identity [68,69] (Supplementary table S3).”
Supplementary Table S3: List of conversions of ectodermal lineage cells into neurons using chemical cocktail.
|
Starting cell type |
Generate cells |
Chemical cocktail used |
Reference |
|
Müller cells |
Neuron-like cells |
Forskolin, ISX9, CHIR99021, I-BET151, and Y-27632 |
[68] |
|
Apical papilla stem cells |
Neuron-like cells |
VPA, CHIR99021, RepSox, Forskolin, SP600125, GO6983, Y-27632 |
[69] |
- How much these small molecules penetrate the brain/ cross blood brain barrier were not discussed in this manuscript.
Answer:
We acknowledge reviewer’s concern regarding the penetration of these molecules across the blood brain barrier (BBB). The diffusion of drugs across BBB is a major hurdle in their delivery to the central nervous system and is dependent on their physiochemical and molecular properties like molecular weight, polar surface area, hydrogen bond, lipophilicity, and charge (Mikitsh JL, Chacko A-M. Pathways for Small Molecule Delivery to the Central Nervous System across the Blood-Brain Barrier. Perspectives in Medicinal Chemistry. January 2014. doi:10.4137/PMC.S13384). Indeed, there are reports suggesting that certain small molecules cross the BBB via passive diffusion.
With regard to the molecular weight, reports suggest that molecules with molecular weight under 400 Da to 500 Da show significant diffusion across the BBB (Misra, Ambikanandan, et al. "Drug delivery to the central nervous system: a review." J Pharm Pharm Sci 6.2 (2003): 252-273.), (Levin, V.A., Patlak, C.S. & Landahl, H.D. Heuristic modeling of drug delivery to malignant brain tumors. Journal of Pharmacokinetics and Biopharmaceutics 8, 257–296 (1980). https://doi.org/10.1007/BF01059646). This diffusion of the molecules has been attributed to the transient pores present in the plasma membrane formed by the kinks of unsaturated fatty acid tails (https://doi.org/10.1007/BF02431971). Given that all the molecules we have used in our cocktail have molecular weight under 500 Da, we hypothesis that they can diffuse across the BBB. However, since the diffusion of these small molecules also depend on other factors, there is a need to test their ability to penetrate BBB in an in vitro BBB model.
Accordingly, we have mentioned about the penetration of small molecules across the blood brain barrier in the introduction section.
(Page-2, lines 60-65, red font)
“To bypass any possible side effects associated with transcription factor mediated re-programming, the use of small molecules is increasingly popular [6, 27-29]. This is due to their target-specific effects, and the ease of controlling this process. Moreover, reports suggest that small molecules weighing under 500 Da show significant diffusion across the blood brain barrier, although this also depends on other factors like polar surface area, hydrogen bond, lipophilicity, and charge [30-32].”

Round 2
Reviewer 2 Report
Authors have address most of the comments. I would suggest, this manuscript can be considered for publication in the present form.